# Are Homemade and Commercial Infant Foods Different? A Nutritional Profile and Food Variety Analysis in Spain

**DOI:** 10.3390/nu13030777

**Published:** 2021-02-27

**Authors:** Maria Jose Bernal, Sergio Roman, Michelle Klerks, Juan Francisco Haro-Vicente, Luis Manuel Sanchez-Siles

**Affiliations:** 1Research and Nutrition Department, Hero Group, 30820 Murcia, Spain; mjose.bernal@hero.es (M.J.B.); michelle.klerks@hero.es (M.K.); jfrancisco.haro@hero.es (J.F.H.-V.); 2Institute for Research and Nutrition, Hero Group, 5600 Lenzburg, Switzerland; 3Marketing Department, Facultad de Economía y Empresa, University of Murcia, 30100 Murcia, Spain; sroman@um.es

**Keywords:** Infants, young children, homemade food, commercial infant food, nutritional profile

## Abstract

Complementary feeding (CF) is an important determinant of early and later life nutrition with great implications for the health status and the development of an adequate growth. Parents can choose between homemade foods (HMFs) and/or commercial infant foods (CIFs). There is no consistent evidence as to whether HMFs provide a better nutritional profile and variety over CIFs. The aim of this study was to compare the nutritional profiles and food variety of HMFs versus CIFs in the Spanish market targeted for infants (6–11 months) and young children (12–18 months). Thirty mothers with their children aged 6 to 18 months were included in this cross-sectional study, following a 3-day weighed food diary of which HMFs were collected and chemically analyzed. HMFs meals for infant provided significantly lower energy, higher protein and higher fiber, for young children provided significantly higher protein and fiber than CIFs meals. HMFs fruit purees for infant shown significantly higher fiber and for young children provided higher energy than CIFs. HMFs meals contained a significantly greater number of different vegetables than CIFs meals (3.7 vs. 3.3), with carrot as the most frequently used in both. However, in CIFs fruit purees shown higher different fruits than HMFs, in both the banana was the fruit most frequently used. There was a predominance of meat and lack of oily fish and legumes in both HMFs and CIFs meals. HMFs and CIFs were equally characterized by a soft texture and yellow-orange colours. Importantly, our findings emphasize the need for clear guidelines for the preparation of HMFs as well as the promotion of food variety (taste and textures) in both HMFs and CIFs to suit infants’ and young children’s nutritional and developmental needs.

## 1. Introduction

Complementary feeding (CF) is an important determinant of early and later life nutrition with great implications for the health status and the development of an adequate growth [1,2]. The feeding transition from breast milk to solid foods is suggested to optimally start at 6 months of age [3]. At this moment breast milk is no longer enough to exclusively support infant’s nutrition [2,3,4]. CF presents several challenges from a nutritional point of view [2,5,6,7]. For instance, concerns about energy and nutrient intakes during infancy have shifted from the risk of underweight to excessive intakes that may lead to later childhood obesity [8,9]. This shift has been evidenced in several studies focused on energy and protein overconsumption in infants [10] and increased BMI in childhood [11,12].

When deciding what type of complementary foods to provide to their children, parents have the option to choose between homemade foods (HMFs) and/or commercial infant foods (CIFs). Available worldwide healthcare guidelines favor the use of HMFs over CIFs [7,13,14,15]. In fact, parents generally see HMFs as the preferred, most controlled option to provide fresh, tasty, and nutritious ingredients to their children [16]. However, although in some countries national guidelines and reports for the preparation of HMF exist [17,18,19,20], in general there is a lack of consistent and detailed international guidelines for the preparation of HMFs [21,22], including the quantities that parents should add to the recipe, which poses a challenge to meet an adequate nutrient balance. Conversely, CIFs follow strict regulations based on age-specific dietary recommendations and ensure compliance with the strictest legal requirements on pesticides and contaminants (nitrates, mycotoxins, metals, among other) [23,24].

Furthermore, despite parent’s perceptions, there is no agreement on whether HMFs are nutrition-ally superior to CIFs or vice versa [7,13,14,25,26,27,28]. One of the reasons is that little is known about the exact nutrient composition of HMFs, with only a couple of studies that have investigated it through chemical analyses [29,30]. Despite their valuable insights, these studies are focused on HMFs and do not compare them to CIFs. Furthermore, van den Boom et al.’s (1997) [29] analysis was conducted more than 20 years ago and was restricted to 50 beef meals in Spain, whereas Abeshu et al.’s (2016) [30] findings are limited to selected food insecure woredas in Ethiopia. 

Most of the available research on the nutritional composition of HMFs, compared to CIFs has been carried out through dietary records, cookbooks recipes and online surveys [24,25]. The reliability of this evidence against real composition is questionable. For example, parents may modify the recipes based on personal preferences or cultural backgrounds. Additionally, they may omit or mis-calculate the actual amounts of ingredients added in the dietary records [31,32,33].

Additionally, the CF period is a key starting point to expose infants to new tastes and textures that will shape preferences and later acceptance to a wide variety of familiar foods [34,35,36]. Increased levels of food variety maximize the nutrient variety which leads to a complete, balanced diet [37,38,39]. Moreover, there is evidence that low food diversity during the first year of life might in-crease the risk of asthma and allergies in childhood [40]. Despite its relevance, there is no consensus on whether HMFs or CIFs provide a wider food variety [7]. Hypothetically, CIFs, as compared to HMFs, offer a narrower diversity of tastes and textures due to the standardized methods by which they are produced [41]. However, findings from Mesh et al. (2014) [42] in Germany and Carstairs et al. (2016) [27] in the UK indicates that CIFs provide a greater vegetable variety than the HMFs cooked recipes. 

In short, despite its importance, the nutrient composition and food variety in HMFs and CIFs remains virtually unexplored. Therefore, the aim of this study was to compare the nutritional profiles and food variety between HMFs (chemically analyzed) and CIFs available in the Spanish market for infants (6–11 months) and young children (12–18 months). Importantly, our analysis included meat, fish, and vegetable meals as well as fruits purees. Our findings will evidence the deficient or excessive nutritional composition from HMFs and CIFs, thus leading towards the establishment of clear guidelines for the preparation of HMFs, improvements in the design of CIFs, as well as the promotion of food variety (flavors and textures) in HMFs and CIFs to suit infants’ and young children’s nutritional and developmental needs. In the next sections we de-scribe our methodology and present the main findings and its implications.

## 2. Materials and Methods

### 2.1. Study Design

This study is based on data from the Hero Infant Diet and Nutritional Survey (HIDANS), a cross-sectional observational study that included thirty mothers with their children between 6–18 months of age of four Spanish cities (Madrid, Barcelona, Sevilla, and Valencia). Data were registered in October 2014 over a 3-day period. The study was conducted following the Declaration of Helsinki guidelines. The protocol was approved by the Ethic Committee of the University of Murcia and a written informed parental consent was obtained from each participant before inclusion.

### 2.2. Eligible Criteria

Participants were recruited via an independent market research firm, which also assisted in the food collection process. Participants were selected from the firm’s consumer panel. The following screening criteria were established. Eligible subjects consisted of mothers who had at least one child aged 6–18 months; had primary responsibility for their child’s feeding; prepared HMFs at least one day over the three days of the study; their child had a birth weight between 2500–4200 g and did not have severe food allergies or chronic medical problems affecting their food intake. Additionally, quota sampling was used to have a similar representation of the children age across the four cities.

### 2.3. Study Procedure

Mothers were asked to cook and feed their children as they would normally do; that is to say, without changing their habits and routines. They were trained to fill in a 3-day weighed food diary (two consecutive weekdays and one weekend day), take photos of each HMF, and collect all the HMFs samples (see original data provided by parents in the Appendix A). Mothers were asked to report a full description of the HMF recipes; they weighted each ingredient with the provided electronic weighing scale and reported the amounts used. In addition, the date and the time of preparation and the ingested amount of all foods were reported. Recipes were prepared when mothers decided to do so, often this was just before giving the food to their babies, but in some cases, foods had been prepared on a previous day and stored in the freezer (see full description in the Appendix A). Mothers were asked to prepare enough food to feed their babies and to fill two sterile containers of 120 g. These containers were collected and frozen immediately per sample at home and stored until they were taken to a central collection site. All the samples were then kept at −18 °C until analyzed in laboratories at Hero Spain, S.A. The dietary records (*n* = 90) collected were used to evaluate the type and frequency of the food, calculate the prepared and ingested amount, the percentage of adherence to the nutritional recommendations and assess the variability of the ingredients employed by the mothers for the preparation of HMFs.

#### 2.3.1. Chemical Analysis of Nutrient Composition of HMFs Samples

The HMFs samples were analyzed in triplicate for each nutrient determination. The procedures established by the Association of Official Analytical Chemists (AOAC) were followed in this step. Protein content was evaluated using the micro-Kjeldahl method, applying a conversion factor (F) of 6.25, based on the method 955.04 [43]. Fat content was evaluated by methods 922.06 and 983.23 [43]. Fatty acids were extracted followed by a dilution in hexane [44] and later with methyl esters formation with methanolic potash 2N and determination by gas chromatography (796/2002 EC). 

Carbohydrate determination was performed as total carbohydrate by difference [45]. Results from the individual analysis of fat (F), ash (A), protein (P), dietary fiber (DF) and water (W) were subtracted from the total weight of the food to obtain the amount of carbohydrates. That is to say: carbohydrates (%) = 100 − (F + A + P + DF + W). 

Sugar profile (sucrose, maltose, glucose and lactose) was determined by High-performance liquid chromatography [46]. Total dietary fiber was determined by the enzymatic-gravimetric methods described by Prosky et al. (1985) [47] (Method 991.43 section 32.1.17 AOAC 1994) [48] with The Fibertec™ 1023 incubation and filtration system. 

Energy content was determined by the Atwater conversion factors [45], by which protein and carbohydrate are multiplied by 4 kcal (12 kJ) and fat by 9 kcal (37 kJ). Sodium content was deter-mined by ash analysis 923.03 [43], followed by atomic absorption spectrophotometry (973.54) [43,49].

The percentage of recommendation of each nutrient in HMFs and CIFs were calculated based on EFSA (European Food Safety Authority) for infant (between 6 to 11 months) and young children (between 12 and 18 months) [5]. Specifically, we took into account the ingested portion size of each recipe of HMFs and the portion size of CIFs, as sold in the Spanish market.

#### 2.3.2. CIFs in the Spanish Market

A comparative analysis of the nutritional profile was performed between the HMFs samples analyzed and collected and CIFs from the four main manufacturers in the Spanish market. For this comparative analysis, we chose those complementary food semi-solid, pureed, and mashed foods offered as meals (containing meat, fish or vegetables) and fruit purees. Data of the CIFs was collected from August 2016 to April 2017 from the major market information suppliers (Nielsen and IRI). All CIFs data was included in a web-based database and the variables followed a similar protocol to previous publications [50,51]. The variables entered included: brand, product name, food type, recommended age, portion size, ingredients, and nutritional content. The nutritional information and ingredients data of the CIFs was obtained from information shown in the labels available on the manufactures’ websites. Inclusion criteria for the database consisted of CIFs meals and fruit purees targeted to infants of 6 months and older, so as to be consistent to the HMFs samples collected. More than 90% of the available products were included in order to have a significant representation of the CIFs in the Spanish market. Others complementary food as infant cereals, fruit juices, snacks, and finger foods were not included in the analysis. 

As for the sample of CIFs in the Spanish market (*n* = 143), 73% consisted of meals (74% meat, 19% fish and 7% vegetables) and 27% corresponded to fruit purees. Of the total of 143 CIFs, 86% were for infants below 1 year, whereas only 14% were available for young children of 12 months on-wards.

#### 2.3.3. Food Variety

Food variety was evaluated and described for HMFs recipes and CIFs of the Spanish marker in meals and fruit purees. Frequencies of their ingredients were calculated. Additionally, the total number of vegetables and fruits included, per meal or fruit puree, was calculated and compared between HMFs and CIFs [27]. Frequencies of the vegetables and fruits present as main vegetable or fruit ingredient in the label were calculated for the CIFs. Likewise, for HMFs, frequencies were calculated for the vegetables and fruits with the highest amount in the recipes as reported by the mothers. 

#### 2.3.4. Food Texture and Colour 

A descriptive evaluation of the food texture and colour was performed on HMFs and CIFs. Photos of the HMFs, registered in the 3-day dietary records and CIFs photos from the manufactures’ websites were visually evaluated. The evaluation was independently conducted by two researchers, and any disagreement was discussed and resolved with a third researcher. Both types of complementary food were classified into three food textures (smooth, smooth with small pieces and minced/chopped) [13,14,52,53]. Colour classification included five colours (yellow, orange, green, red and purple/blue) and “others” that corresponded to food with a mix of colours, characteristic of the minced/chopped products. Frequencies were calculated for both texture and colour classifications of the two types of complementary food (HMFs and CIFs from market).

### 2.4. Statistical Analysis

All analyses were conducted with the use of the Statistical Package for the Social Sciences (SPSS) v.19.0 (IBM SPSS Inc., Chicago, IL, USA). Data was checked for normality by Shapiro-Wilk Test. Overall, nutritional data was found to be not normally distributed (unless stated otherwise) and therefore analyzed through a non-parametric test. Mann-Whitney U test was used to evaluate the differences between the nutrient composition of HMFs and CIFs. The food variety data was normally distributed and the differences between HMFs and CIFs were analyzed through Independent t-test. Non-parametric data was presented in medians and interquartile range (Q1, Q3), while normal data as means ± standard deviation (SD). Since there is no established rule regarding the difference needed to establish a nutritional significance, effect sizes were calculated (r). Statistical significance was set a *p* value < 0.05.

## 3. Results

### 3.1. Sample Characteristics

Our sample consisted of 30 children grouped according to age in infants of 6–11 months and young children of 12–18 months. Population characteristics are displayed in Table 1. 

Total complementary food intake was on average higher in young children than in infants (690.0 ± 243.0 vs. 465.6 ± 185.5 g/day, *p* = 0.01). Conversely, young children consumed, on average, less breast milk/formula than infants (350.9 ± 234.5 vs. 677.4 ± 228.0 ml/day, *p* ≤ 0.001). Almost all infants and young children consumed commercial infant cereals (94% and 85%, respectively) on at least two of the three days of the study. Infants consumed significantly more commercial infant cereals than young children (*p* = 0.002) (Table 1).

### 3.2. Frequency and Type of HMFs and CIFs Consumed 

A total of 121 recipes of HMFs, including meals (71%) and fruits purees (29%), were prepared by the mothers in our study. The HMFs meals were classified as meat (56%), fish (34%) or only vegetables (10%). It was observed that 76% of the infants and 69% of the young children consumed HMFs the three-day study. Meat and fish meals were the most consumed HMFs meals with 82% of the infants and 92% of the young children that ate at least one time per day such meals, at least one day of the study. Additionally, 18% of the infants and 31% of the young children consumed these meals two times per day, at least one day of the study.

Additionally, 43% of the total children also consumed CIFs: 36% meals (30% fish and 70% meat) and 64% fruit purees. Consumption of CIFs during the three-day study was observed in 18% of the infants and 8% of the young children. In general, children ingested more CIFs fruit purees (33% of children) than meat or fish meals (26% children). Particularity, the percentage of infant and young children who consumed at least one day of the three-day study meals CIFs was 52% and 30%, respectively (Table 2). The total consumption of HMFs for both fruit purees and meals, was higher than CIFs, for both age groups. However, this higher consumption of HMFs vs. CIFs was only significant in meals for young children (*p* = 0.042) (Table 2).

### 3.3. Portion Size, Nutrient and Energy Profile Comparison between HMFs and CIFs 

#### 3.3.1. Meals 

The main differences between HMFs and CIFs were found for portion size, energy, protein, and fiber. Median portion size was significantly higher in CIFs than in HMFs for both age groups. In particular, median portion size was 200.0 g vs. 185.5 g, respectively (*p* ≤ 0.001; *r* = 0.6) and for young children, median portion size was 250.0 g vs. 218.0 g, respectively (*p* = 0.04; *r* = 0.3) (Table 3). Age differences in portion size were only found for CIFs with higher portion size in the meals for young children (*p* ≤ 0.001; *r* = 0.3). 

Median energy density (ED) was higher in CIFs compared to HMFs. HMFs for infants showed the lowest ED (48.0 kcal/100g, *p* ≤ 0.001; *r* = 1.2), while CIFs contained 43% more energy. Age differences were noted for energy with higher energy in HMFs for young children, compared to infants (66.2 vs. 48.0 kcal/100g, *p* ≤ 0.001; *r* = 0.5) (Table 3), however in CIFs no significant differences were found in ED between ages (68.5 vs. 69.5 kcal/100 g, in infant and young children respectively, *p* = 0.37).

Median protein content was higher in HMFs than in CIFs for both age groups. More specifically, median protein was 3.6 vs. 3.0 g/100 g, (*p* = 0.002, *r* = 0.4) for infants and 3.8 vs. 2.9 g/100g, (*p* ≤ 0.002, *r* = 0.5) for young children. Median fat content was higher in CIFs only for infants (2.3 vs. 1.5 g/100 g, *p* ≤ 0.001; r ≥ 0.8). Median fiber content was higher in HMFs than CIFs for both age groups. For infants, median fiber was 2.6 vs. 0.8 g/100 g, respectively (*p* ≤ 0.001; *r* = 1.3) and for young children 2.0 vs. 0.8 g/100 g, respectively (*p* ≤ 0.001; *r* = 0.7). Median sodium content was similar in HMFs and CIFs for infants, while it was almost double in CIFs compared to HMFs for young children (111.0 vs. 65.2 mg/100 g; *p* = 0.04; *r* = 0.3) (Table 3).

#### 3.3.2. Fruits Purees

As shown in Table 3, differences were mainly found in portion size, energy, carbohydrate content and fiber between HMFs and CIFs fruit purees. Median portion size was significantly higher in HMFs than in CIFs, but only for young children (206.0 vs. 100.0 g, *p* ≤ 0.001; *r* = 0.8).

Highest median ED was found in the HMFs fruit purees for young children (78.0 kcal/100 g). This was significantly higher than in the CIFs counterparts (61.0 kcal/100g; *p* = 0.02; *r* = 0.5). Age differences for energy were found with the CIFs for infants being more energy dense than the ones for young children (72.0 vs. 61.0 kcal/100 g, *p* ≤ 0.001; *r* = 0.6). Despite the difference in carbohydrate content between HMFs and CIFs for young children (16.3 vs. 13.0 g/100 g; *p* = 0.03; *r* = 0.3), no differences were found in sugar content, either between the type (HMFs or CIFs) or between both age groups. Consistent to the pattern in HMFs meals, fiber content was higher in HMFs fruit purees than in CIFs for both age groups, but only significant for infants (*p* ≤ 0.001; *r* = 0.7). 

### 3.4. Food Variety

The overall food variety included in HMFs and CIFs is described for both meals and fruit purees in Figure 1. A small percentage of meals containing only vegetables (without meat or fish), was present in both HMFs and CIFs meals (≤10%). A similar total number of different vegetables was used in HMFs and CIFs meals (15 vs.14) (Table 4). The most frequently used vegetable in HMFs meals, independently of the amount included, was carrot (78%), and it was the second in CIFs meals (82%) after onion (86%) (See Figure 1).

The frequency of vegetables and fruit used in HMFs and CIFs, by age group, are shown in Table 4. Similar ingredients were found in meat and fish meals for HMFs and CIFs, with the predominance of chicken in meat meals (67% and 41%, respectively) and hake in fish meals (69% and 65%, respectively).

Other ingredients, such as potato were included in the majority of HMFs and CIFs meals (91% and 61%, respectively). Salt addition was reported in 42% of the total HMFs meals (36% for infants and 64% for young children); whereas less than 10% of CIFs contained added salt, as declared in the label and only for the meals intended for young children. Interestingly, of the total HMFs vegetable meals (*n* = 9), 89% contained added salt, while 27% of the total HMFs fish meals (*n* = 29) and 42% of the total HMFs meat meals (*n* = 48) had added salt (Figure 1). 

Regarding the number of different vegetables per meal, HMFs meals contained significantly more vegetables than CIFs meals (3.3 vs. 3.7, *p* = 0.012). As for fruit purees, total number of different fruits is more than double in CIFs as compared to HMFs (15 vs. 7, respectively). Still, in both HMFs and CIFs fruit purees, the most frequently used fruit was banana (91% vs. 95%, respectively) (Table 4). However, CIFs and HMFs fruit purees contained the same mean number of different fruits per puree (3.3). CIFs fruit purees were added with sugar in the form of juice concentrate in only 23% of them.

### 3.5. Food Texture and Colours in HMFs and CIFs

The texture of HMFs meals and fruits was mainly smooth (95%) and only 5% of the HMFs (meals only) consisted of a minced/chopped texture (See Figure 2). The minced/chopped HMFs found corresponded only to 10% of the children included in the study (one infant of 6–11 months and two young children of 12–18 months). Similarly, for CIFs meals and fruits, mainly smooth textures (92%) were offered, whereas, only 1% was offered as minced (meals only). The remaining CIFs (7%) had a smooth texture with small pieces added of meat and/or vegetables.

Regarding food colour, HMFs were predominantly yellow (67%), followed by orange (20%), green (7%) and others (6%, only in meals with a minced/chopped texture). On the other hand, more than half of CIFs were orange (61%), followed by yellow (36%). Only 2% had a red colour and the remaining 1% had other colours.

## 4. Discussion

Whether to give HMFs, CIFs or a combination of them represents a challenge as parents want to provide the best nutrition quality to their children during the CF period. This study compared the nutritional composition of 121 HMFs (meals and fruit purees) with 143 CIFs (meals and fruit purees) for infants and young children and described the food variety in HMFs and CIFs in Spain. A major strength in our research was to chemically analyze a wide spectrum of the actual HMFs prepared by 30 mothers from 4 different cities without any intervention to evaluate their nutrient profile. Several implications can be derived from our study. 

### 4.1. Nutritional Profile Comparison 

The most remarkable nutritional differences found in our study between HMFs and CIFs meals were: (1) the lower ED in HMFs compared with CIFs for infants, (2) the higher protein content in HMFs compared to CIFs and (3) the lower fiber content in CIFs compared to HMFs for both age groups. Nevertheless, in general the ED found in our samples of HMFs and CIFs meals were higher than the recommended minimum threshold (60 kcal/100 g) by WHO (2019b), except for HMFs meals targeted at infants, where the ED was 48 kcal/100 g. This can be explained by the low contribution of fat to the percentage of total energy in HMFs meals for infant studied (28%E) compared to the recommendations (40%E at 6–11 months of age and 35–40% at 12–18 months of age) [5]. However, the mean percentage of energy from fat in CIFs was also low in both infant (30%E) and young children (32%E). Surprisingly, the intake of fat is somehow underestimated because essential fatty acids (linoleic and α-linolenic acid, as dietary precursors of arachidonic and docosahexaenoic acid) play a key role in brain development of infant and young children [2,5]. 

Overall, we found an imbalance of macronutrient content in HMFs meals for infants, with a high proportion of protein (30%E), whereas fat was considered to be low (28%E). These results are consistent to van den Boom et al. (1997), who found a low ED (50 kcal/100 g), a low-fat content (25%E) and a high protein content (35.2%E) [29].

Despite HMFs and CIFs meals had a protein content within the minimum values established by the European legislation (3 g/100 kcal) [23], it is important to mention the risk of exceeding the daily recommendations for protein of ≤11 g/day for infants and ≤13 g/day for young children [5]. Our study evidenced a significantly higher protein content in HMFs, as compared to CIFs. This is particularly relevant if we consider the portion size and daily frequency of HMFs meals, as well as other type of foods given to children on a daily basis. For example, an infant having a HMF meal, with a portion size of 185.5 g (which was common in our study), would be consuming 6.7 g of protein only from that meal, which represents 61% of the daily adequacy for protein. This intake along with breast milk/formula intake and other protein-rich foods included in the daily diet would easily exceed the recommended values. Unfortunately, this is in line with data from the European Childhood Obesity Project that evidenced Spain (together with Italy) with the highest protein intakes, exceeding the recommended 15%E coming from protein in 90% of the children younger than 2 years [54]. This is of particular concern since the latest position of the European Society for Paediatric Gastroenterology, Hepatology, and Nutrition on CF stressed the association found between higher protein intake, during infancy, and the risk of a higher BMI during childhood [2,55]. 

In addition, HMFs had higher fiber content as compared to CIFs in our study. This might be explained by the industrial process (heat treatment and mechanical process) where a portion of fiber is likely be lost [56,57]. It is important for manufacturers to optimize the recipe’s design and the industrial process for achieving higher contents of fiber in CIFs. Notably, fiber plays a relevant role in the prevention of several diseases such as cardiovascular diseases, type 2 diabetes, obesity, or cancer [58], and promotes gut microbiota development in infant and young children [59,60]. In addition, despite the fact that there are no fiber recommendations until 1 year of age [5], pediatric organizations generally recommend the consumption of foods rich in fiber such as legumes, vegetables, fruits and whole grains from the beginning of CF [13,14,53,61]. 

Regardless of the recommendation of avoiding salt addition to the complementary meals [5,52,62] a high proportion of HMFs meals were prepared with salt (42%), a pinch as reported by the mothers, compared to CIFs (<10%). Interestingly, the addition of salt was predominantly for vegetable meals since mothers may add salt in such cases so as to “mask” the characteristic bitter taste of vegetables and thus hoping to increase their acceptance. However, findings from Bouhlal, Issanchou, Chabanet, and Nicklaus (2014) [63] showed that such procedure is not as effective as repeated exposure to vegetables (without added salt). Even though, our results show that sodium values for both HMFs and CIFs meals were within the established European limits (<200 mg/100 g) [21], giving meals to infants with added salt at early ages might condition their preference for salty tastes later in life [64,65]. 

On a related issue, even though we found no significant differences in sugar content between HMFs (<12.6 g/100g) and CIFs fruit purees (<11.7 g/100 g); both exceed 10 g/100 g, which can be considered too be high [66]. One plausible reason is the frequent use of banana (more than 90% in both recipes). Thus, we suggest increasing the use of other fruits less sweet and more acid in both HMF and CIFs for stimulating the acceptability of a wider range of fruits.

### 4.2. Food Variety, Texture and Colour Comparison

HMFs meals in the current study that contained vegetables had an average number of different vegetables (3.7 ± 1.3) significantly higher than CIFs (3.3 ± 1.1) and HMFs from the DONALD study in Germany (1.6 ± 1.3) [38]. Even though the use of carrot as the main vegetable ingredient was in line with previous studies [27,42,51,67], the use of green vegetables in HMF and CIFs in our study was higher as compared to those found in Germany [42] and UK [27]. Still, the use of these green vegetables is limited both in HMFs and CIFs. Most likely, this is to avoid food rejection. Given the benefits of green vegetables we encourage both parents and manufacturers to increase the use of them. 

Overall, meat meals were more frequent than fish meals, which is consistent to previous studies [7,42]. However, contrary to findings from Mesch et al., (2014) [42] in Germany, fish meals in our study were higher and more frequent in HMFs as compared to CIFs. One plausible explanation is that Spain is one of the largest consumers of fish in Europe, whereas Germany is among the smallest consumers (EU Fish Market 2019). On a related issue, despite the health and growth benefits of DHA-rich fish in infants such as salmon or tuna [68,69], its use in HMFs and CIFs meals was absent in our study. 

Consistent to prior studies in Spain [70], the use of legumes was limited in both HMFs (9%) and CIFs (1%) meals. Given its key role in nutrition in general, its use in both HMFs and CIFs should be encouraged.

Food texture is another aspect of relevance when investigating food variety [71,72]. We found in our samples of HMFs and CIFs a considerably low texture variety, with the predominance of smooth textures. This result has also been found in Australian CIFs (*n* = 84 products), where only 23% of the product were chunky [73]. Accordingly, we encourage both industry and parents to use a wider spectrum of textures (e.g., lumpy or mashed foods) in order to promote infants and young children to develop mastication and adapt to familiar foods (with more complex textures and tastes). Research has found that this effort is particularly important when 9–12 month infants are in their transition towards chopped/minced foods [2,35,52].

Finally, our research explored the colour of HMFs and CIFs meals, which represents an additional aspect of food variety. The colour of foods is an indirect way to measure the ingredient variety and consequently nutrient variety of foods [74]. Prior evidence shows that the colour of fruits and vegetables is related to their specific antioxidant and phytochemical content [75]. Leading health organizations have encouraged that infants eat fruits and vegetables of different colours, which stimulate a balanced diet and food acceptance [2]. The two main colours found in HMFs and CIFs meals were yellow and orange, which are associated with the presence of carotenoids [76] and correspond to the frequent use of carrots. Regardless of the use of green vegetables, such as zucchini, the fact that vegetables were peeled as recommended [52] explains, to some extent, the limited presence of green colour in HMFs meals. 

### 4.3. Limitations and Future Research

We believe our results provide relevant insights related to the nutrition of infants and young children, yet we acknowledge that there are some limitations that represent the foundation of additional work. For example, although we considered a wider age range of children (6–18 months) in four different Spanish cities and different educational backgrounds of the mothers, our findings are restricted to Spain. Given the importance of culture in foods, we encourage similar studies to be developed in other countries. The number of home-made recipes analyzed in our study is similar to that of previous studies [27,76,77,78]. Collecting and chemically analyze 121 home-cooked meals from 30 mothers in four different cities was extremely complex from a logistic point of view, but also very expensive and time consuming. Nevertheless, future studies would benefit from using a larger sample of home-made recipes. In addition, we compared HMFs and CIFs from two different periods (2014 and 2016/2017 respectively). Our comparison is valid and sound as infant and toddler foods are highly regulated in Europe by the Commission Directive 2006/125/EC [23]. Rules are especially strict with minimum and/or maximum limits for different nutrients. Importantly, the Commission Directive 2006/125/EC was legally binding in both periods of data gathering (2014; 2016–2017), which implies that the margin to modify nutrients and substances in infant meals and fruit purees is limited as shown in previous studies [79].

Future research is needed to examine energy and nutrient intakes from the complete daily diet (i.e., milk, infant cereals, and other types of foods). Information regarding CIFs meals was obtained from the labels and manufacturers’ site and future research could be done on the chemical analysis of both the HMFs and CIFs. Finally, texture and colour variety analysis was implemented through visual, subjective evaluations.

## 5. Conclusions

To the best of our knowledge this is the first study that has compared the nutritional profile of HMFs (chemically analyzed) to CIFs, along with an evaluation of food variety, texture and colour. Although no overall superiority of HMFs over CIFs or vice versa was found in this study, our findings stress the need to establish clearer guidelines on the preparation of HMFs, with exact amounts and proportions of all food sources to add to the recipes in order to avoid imbalanced diets that may lead to future health risks. Moreover, we encourage R+D departments of infant food manufacturers to lead the improvement of CIFs through the optimization of recipes and the provision of higher ingredient variability.

## Figures and Tables

**Figure 1 nutrients-13-00777-f001:**
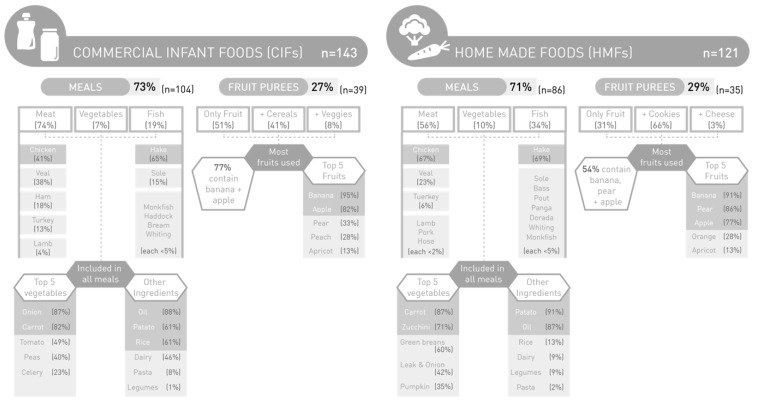
Food variety in HMFs and CIFs meals and fruit purees for infants (6–11 months) and young children (12–18 months). (Percentages of meals and fruit purees containing each of the different ingredients).

**Figure 2 nutrients-13-00777-f002:**
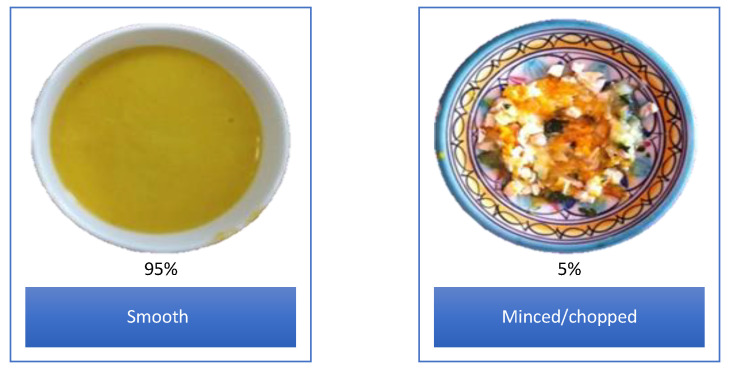
Examples of the HMFs for texture classification. Pictures were taken by the mothers included in this study.

**Table 1 nutrients-13-00777-t001:** Population characteristics.

	6–11 Months (*n* = 17)	12–18 Months (*n* = 13)
*Characteristic of the children*		
Age (months) (mean ± SD)	8.3 ± 1.4	14.4 ± 1.9
Gender (% Male)	58.8	84.6
Birth weight (g) (mean ± SD) ^1^	3173.3 ± 455.0	3178.9 ± 451.3
Actual weight (kg) (mean ± SD)	8.6 ± 1.3	10.6 ± 1.2
Attend day-care (%) ^1^	16.7	68.8
Total complementary food intake (g/day)	465.6 ± 185.5	690.0 ± 243.0 *
Commercial infant cereals intake (g/day)	50.8 ± 27.7	22.5 ± 13.1 *
BM/Formula intake (mL/day)	677.4 ± 228.0	350.9 ± 234.5 *
*Characteristic of the mothers ^1^*		
Age (years) (mean ± SD)	32.5 ± 4.5	35.1 ± 5.2
Education (%)		
Primary school	46.7	12.5
High school	33.3	37.5
Some college	33.3	31.3
Univ. or above	16.7	18.8
Main activity (%)		
Employed	50.0	69.0
Student	8.3	0
Unemployed	41.7	31.0
Family Status (%)		
Single/divorced	8.3	6.3
Married	75.0	62.5
Couple	16.7	31.3

^1^ Data presented corresponds to *n* = 28, due to lack of information provided by 2 of the mothers in the study. * Significance difference between 6–11 months and 12–18 months (*p* < 0.05).

**Table 2 nutrients-13-00777-t002:** Average Daily Intake of Homemade Foods (HMFs) and Commercial Infant Foods (CIFs) from 3-days Weighted Food Diaries.

	6–11 Months (*n* = 17)			12–18 Months (*n* = 13)		
Type of food	HMFs	CIFs	*p* Value	HMFs	CIFs	*p* Value
Number of infant or young children who consume (%)	17 (100%)	9 (53%)		13 (100%)	4 (31%)	
Fruit purees intake (g/day) ^1^	106.1 ± 68.9	75.5 ± 34.6	0.20	166.1 ± 99.9	86.7 ± 0	0.50
Meals intake (g/day) ^1^	186.9 ± 115.5	83.3 ± 29.6	0.15	191.1 ± 105.7	70.8 ± 8.3	0.042
Meals (%) ^2^	94	18		100	31	
Meat (%) ^1^	82	12		85	23	
Fish (%) ^1^	65	12		69	8	
Vegetables (%) ^1^	12	-		38	-	
Fruits purees (%) ^1^	70	53		46	8	

^1^ The intake (g/day) refers to the mean amount of Homemade Foods (HMFs) and Commercial Infant Foods (CIFs) consumed per day by infants; ^2^ Percentage of infants and young children consuming each HMFs/CIFs type.

**Table 3 nutrients-13-00777-t003:** Comparison of the nutritional profile (per 100g) of chemically-analyzed HMFs and CIFs for infants (6–11 months) and young children (12–18 months). Values are median (Q1, Q3).

	6–11 Months (*n* = 17)			12–18 Months (*n* = 13)			Overall (*p* Value) ^1^	
	HMFs	CIFs	*p* Value	HMFs	CIFs	*p* Value	HMFs	CIFs
*Meals (n)*	48	94		38	10			
Portion size (g/meal) ^2^	185.5 (138.7, 211.5)	200.0 (200.0, 250.0)	≤0.001	218.0 (113.5, 254.5)	250.0 (246.2, 250.0)	0.04	0.30	≤0.001
Energy (kcal)	48.0 (34.5, 54.6)	68.5 (64.0, 71.0)	≤0.001	66.2 (52.8, 80.0)	69.5 (66.0, 72.0)	0.49	≤0.001	0.37
Protein (g)	3.6 (2.9, 4.3)	3.0 (2.6, 3.6)	0.002	3.8 (3.4, 4.6)	2.9 (2.6, 3.0)	≤0.001	0.21	0.24
Carbohydrates (g)	3.9 (2.9, 5.0)	8.4 (7.6, 9.2)	≤0.001	5.1 (3.6, 6.4)	7.9 (7.3, 9.0)	≤0.001	0.03	0.48
Sugar (g)	1.3 (1.0, 1.5)	1.3 (0.9, 1.8)	0.60	1.4 (0.8, 2.1)	1.3 (1.1, 1.8)	0.85	0.49	0.85
Fiber (g)	2.6 (2.0, 3.0)	0.8 (0.5, 1.0)	≤0.001	2.0 (1.4, 2.8)	0.8 (0.8, 1.0)	≤0.001	0.03	0.15
Fat (g)	1.5 (0.6, 2.1)	2.3 (2.2, 2.5)	≤0.001	2.7 (1.2, 4.5)	2.5 (2.4, 2.7)	0.79	≤0.001	≤0.001
Saturated (g)	0.3 (0.1, 0.35)	0.6 (0.5, 0.7)	≤0.001	0.4 (0.3, 0.7)	0.5 (0.5, 7.2)	0.11	≤0.001	0.68
Sodium (mg)	34.4 (23.1, 56.0)	28.0 (28.0, 48.0)	0.86	65.2 (39.5, 126.1)	111.0 (93.0, 140.0)	0.04	≤0.001	≤0.001
*Fruit purees (n)*	22	29		13	10			
Portion size (g/fruit) ^2^	172.0 (129.5, 214.7)	200.0 (95.0, 250.0)	0.77	206.0 (188.5, 245.0)	100.0 (90.0, 113.0)	≤0.001	0.048	0.01
Energy (kcal)	69.2 (64.7, 71.9)	72.0 (69.5, 78.5)	0.06	78.0 (63.2, 84.8)	61.0 (56.7, 63.0)	0.02	0.06	≤0.001
Protein (g)	1.0 (0.9, 1.2)	0.8 (0.7, 1.0)	0.003	1.0 (0.7, 1.1)	0.8 (0.5, 0.8)	0.02	0.57	0.28
Carbohydrates (g) ^3^	15.3 ± 2.5	16.0 ± 2.3	0.30	16.3 ± 3.3	13.0 ± 0.70	0.005	0.30	≤0.001
Sugar (g) ^3^	12.6 ± 1.4	11.9 ± 2.2	0.26	11.9 ± 1.9	11.7 ± 0.7	0.77	0.22	0.60
Fiber (g) ^3^	2.1 ± 0.2	1.3 ± 0.5	≤0.001	1.9 ± 0.28	1.6 ± 0.48	0.13	0.02	0.11
Fat (g)	0.6 (0.2, 0.7)	0.2 (0.1, 0.4)	0.005	0.6 (0.2, 0.5)	0.2 (0.1, 0.3)	0.02	0.42	0.79
Saturated (g)	0.3 (0.2, 0.3)	-	≤0.001	0.3 (0.1, 0.5)	-	≤0.001	0.69	0.47
Sodium (mg)	21.7 (19.8, 21.7)	12.0 (6.0, 16.0)	≤0.001	21.7 (17.6, 30.1)	12.0 (12.0, 20.0)	0.003	0.57	0.10

^1^ Comparison between the two age groups for HMFs and CIFs. ^2^ The portion size is the median amount ingested by infants per HMF meal according to food diaries vs. the median portion size available of CIFs in the Spanish market. ^3^ Data normally distributed, presented in mean ± SD.

**Table 4 nutrients-13-00777-t004:** Percentage of HMFs and CIFs including different vegetables and fruits in meals and fruit purees for infants (6–11 months) and young children (12–18 months).

	6–11 Months ^1^		12–18 Months ^1^	
Meal purees	HMFs (*n* = 48)	CIFs (*n* = 94)	HMFs (*n* = 38)	CIFs (*n* = 10)
Vegetable type ^1^				
Carrot	64.6 (16.7)	80.8 (52.9)	94.7 (23.7)	90.0 (30.0)
Zucchini	68.7 (47.9)	1.9 (1.0)	73.7 (44.7)	10.0 (-)
Green beans	58.3 (8.3)	22.1 (8.6)	63.1 (10.5)	-
Onion	33.3 (10.4)	79.8 (3.8)	52.6 (13.1)	80.0 (-)
Tomato	27.1 (6.2)	41.3 (15.4)	7.9 (-)	80.0 (70.0)
Leek	47.9 (-)	6.7 (1.9)	34.2 (2.6)	20.0 (-)
Pumpkin	50.0 (10.4)	1.0 (-)	15.8 (5.3)	-
Celery	6.2 (-)	20.2 (-)	2.6 (-)	30.0 (-)
Peas	8.3 (-)	37.5 (6.7)	2.6 (-)	30.0 (-)
Pepper	-	1.9 (-)	5.3 (-)	20.0 (-)
Broccoli	4.2 (2.1)	-	2.6 (-)	10.0 (-)
Chard	-	-	13.1 (-)	
Fennel	-	-	-	10.0 (-)
Turnip	-	-	7.9 (-)	-
Parsnip	-	2.9 (-)	2.6 (-)	-
Lettuce	-		2.6 (-)	
Spinach	-	1.4 (-)	-	-
Fruit purees	HMFs (*n* = 22)	CIFs (*n* = 29)	HMFs (*n* = 13)	CIFs (*n* = 10)
Fruit type ^1^				
Banana	86.4 (4.5)	93.1 (34.5)	100.0 (15.4)	100.0 (40.0)
Apple	72.7 (22.7)	86.2 (44.8)	84.6 (61.5)	70.0 (30.0)
Pear	90.9 (31.8)	27.6 (6.9)	76.9 (7.7)	50.0 (10.0)
Orange	68.2 (36.4)	10.3 (6.9)	61.5 (7.7)	-
Peach	4.5 (4.5)	31.0 (3.4)	-	20.0 (-)
Mandarin	4.5 (-)	10.3 (3.4)	7.7 (7.7)	-
Apricot	-	10.3 (-)	-	20.0 (-)
Kiwi	-	3.4 (-)	-	20.0 (-)
Strawberry	-	-	-	20.0 (-)
Plum	-	3.4 (-)	15.4 (-)	-
Mango	-	6.9 (-)	-	10.0 (-)
Pineapple	-	3.4 (-)	-	10.0 (-)
Passion fruit	-	-	-	10.0 (-)
Raspberry	-	3.4 (-)	-	-
Cranberry	-	3.4 (-)	-	-

^1^ Vegetables and fruits are shown in descendent order according to their contribution in the HMFs and CIFs meals and fruit purees, respectively. Results expressed in parentheses correspond to the percentage of meals/fruit purees where the vegetable/fruit type is mentioned as the main ingredient in the recipe.

## Data Availability

The data presented in this study are available on request from the corresponding author.

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
