# Peer review of "Are Homemade and Commercial Infant Foods Different? A Nutritional Profile and Food Variety Analysis in Spain"

_nutrients, 2021, doi:10.3390/nu13030777_

Round 1

Reviewer 1 Report

This is a very relevant study conducted by Bernal et al. The authors performed an outstanding work and I have no objection in accepting this manuscript in its presente form. The introduction provide sufficient background and include all relevant references; The research design is appropriate; The methods are adequately described; the results are clearly presented and discussed and the conclusions are supported by the results. The authors also pointed out the study limitations properly.

Author Response

This is a very relevant study conducted by Bernal et al. The authors performed an outstanding work and I have no objection in accepting this manuscript in its presente form. The introduction provide sufficient background and include all relevant references; The research design is appropriate; The methods are adequately described; the results are clearly presented and discussed and the conclusions are supported by the results. The authors also pointed out the study limitations properly.

We are extremely glad that you liked the paper. We do appreciate the time you have spent reviewing our manuscript and your encouragement. Thank you very much for your positive comments!

Reviewer 2 Report

The work could be interesting and bring a new perspective on the nutrition of infants and young children, but it contains a lot of methodological errors.

line 49-50 - what does it mean that there is a lack of clear and detailed guidelines for the preparation of HMFs? Polish guidelines include a detailed guide for parents of children aged 0-12 months on how to prepare meals for infants. check what are the guidelines in other countries

line 101 - why was such a small, completely unrepresentative sample of mothers selected? Since the respondents were recruited via an independent market research companies, why were not more women recruited?

line 112-113 - Have two sterile 120 g packages been taken for each meal HMF prepared at home? How many meals did the parents prepare at one time? please describe it in more detail.

line 147-148 Data of the CIFs was collected from August 2016 to April 2017 from the major market information suppliers and data were registered in October 2014 over a 3-day period.  Why is there such a discrepancy in the dates? product composition cannot be compared 2 years after data collection.

line 177 -Color classification - there is no explanation as to why the authors studied color classification, the color does not matter for the nutritional value of the products.

line 225-226 - this higher consumption of HMFs vs. CIFs was only significant in meals for  young children (table 2) - in both age groups (according to Table 2) HMF consumption was higher than CIF, but there was no statistical significance for the results obtained. It seems strange, are you sure the statistical analysis was done correctly?

line 233 - Median portion size was significantly higher in CIFs than in HMFs for both age groups. Median portion size was significantly higher in CIFs than in HMFs for both age groups. These results are in contradiction to those presented in Table 2. The authors present unclear results which require improvement and explanation.

Table 2 - 166.1 ± 99.9 vs 86.7±0  p=0.50, the result is not statistically significant and line 234 - s 200.0 g vs. 185.5 g, respectively (p ≤ 0.001);  the statistical analysis is incorrectly performed or the test was wrongly selected for the analyzed results.

line 255-256 - Median portion size 255 was significantly higher in HMFs than in CIFs, but only for young children (206.0 vs. 100.0  g, p ≤ 0.001;). what are these results? how does the data from table 2 relate?

line 308- 312 - As for fruit purees, total number of different fruits is more than double in CIFs as compared to HMFs. CIFs and HMFs fruit purees contained the same mean number of different fruits per puree. This information is completely contradictory and is not supported by the results presented in table 4.

line 351 - In the older age group, the difference in ED between HMF and CIF is statistically insignificant, and with a larger number of subjects in the group, it is not.

line 373-376 - the calculation of the daily food ration on the basis of the 3-day food diary is missing. Only then would it be possible to compare whether the amount of protein, fats or carbohydrates in HMF is higher than in CIF, and whether it is relevant to the nutrition of infants and young children. The results obtained in this way without reference to the daily food ration are irrelevant.

Reviewer 3 Report

Dear Authors,

Very interesting research. They concern a very important group of infants and young children.
In keywords, remove food variety
Change color to colour throughout the manuscript
In the case of tact and color, they are measured instrumentally. Here we deal with sensory evaluation and it should be described differently in the methodology.

Reviewer
